

# Characterization of the complete mitochondrial genome of the Sunda stink-badger (*Mydaus javanensis*) from the island of Borneo

Vijay Kumar Subbiah[1], Chrishen Robert Gomez[1,2], Dexter Miller Robben[1], Ranjita Subramaniam[1] and Andrew James Hearn[2]

[1] Biotechnology Research Insitute, Universiti Malaysia Sabah, Kota Kinabalu, Sabah, Malaysia
[2] Wildlife Conservation Research Unit, Department of Biology, University of Oxford, Oxfordshire, United Kingdom

## ABSTRACT

**Background**. The Mephitidae is a family of skunks and stink-badgers that includes 12 extant species in four genera, namely, *Mydaus, Conepatus*, *Mephitis* and *Spilogale*. *Mydaus* is the only genus within Mephitidae found outside the American continent, with its distribution limited to the islands of Borneo, Indonesia and Philippines. There are two extant species of *Mydaus i.e.*, *javanensis* and *marchei*. Currently, complete mitogenomes are unavailable for either species. Here, we present the characterization of the first complete mitogenome for the Sunda stink-badger (*Mydaus javanensis*) from the island of Borneo.

**Methods**. Muscle tissue was obtained and the DNA was sequenced using a combination of Illumina Barcode Tagged Sequence (BTSeq) and Sanger sequencing techniques. The genome was annotated with MITOS and manually checked for accuracy. A circular map of the mitogenome was constructed with Proksee. Relative synonymous codon usage (RSCU) and codon frequency were calculated using MEGA-X. The protein coding genes (PCGs) were aligned with reference sequences from GenBank and used for the construction of phylogenetic trees (maximum liklihood (ML) and Bayesian inference (BI)). Additionally, due to the lack of available complete genomes in public databases, we constructed another tree with the *cyt b* gene.

**Results**. The complete circular mitogenome was 16,391 base pairs in length. It comprises the typical 13 protein-coding genes, 22 tRNAs, two ribosomal RNA genes, one control region (CR) and an L-strand replication origin ($O_L$). The G+C content was 38.1% with a clear bias towards A and T nucleotides. Of the 13 PGCs, only ND6 was positioned in the reverse direction, along with five other tRNAs. Five PCGs had incomplete stop codons and rely on post-transcriptional polyadenylation (TAA) for termination. Based on the codon count, Leucine was the most common amino acid (589), followed by Threonine (332) and Isoleucine (325). The ML and BI phylogenetic trees, based on concatenated PCGs and the *cyt b* gene, respectively, correctly clustered the species with other members of the Mephitidae family but were unique enough to set it apart from *Conepatus*, *Mephitis* and *Spilogale*. The results confirm *Mydaus* as a member of the mephitids and the mitogenome will be useful for evolutionary analysis and conservation of the species.

Corresponding author
Vijay Kumar Subbiah,
vijay@ums.edu.my

## INTRODUCTION

Stink-badgers and skunks are grouped under the family Mephitidae, which includes 12 extant species placed in four genera: *Mydaus* (stink-badgers), *Conepatus* (hog-nosed skunks), *Mephitis* (hooded and striped skunks), and *Spilogale* (spotted skunks). Three other genera under Mephitidae (*Brachyprotoma*, *Palaeomephitis,* and *Promephitis*) are extinct and known only through fossil records. The stink-badger *Mydaus* is the only one within Mephitidae found outside the American continent. Its resemblance to badgers led early authors to classify the species under Mustelidae (subfamily: Melinae), but recent molecular evidence has led to its reclassification as a member of Mephitidae (*Dragoo & Honeycutt, 1997*; *Hwang & Larivière, 2003*). To date, the genus includes two species: *Mydaus javanensis* and its only known sister taxon, *Mydaus marchei.*

The Sunda stink-badger (*Mydaus javanensis*), commonly known as the Malay Badger or Teledu, is one of Southeast Asia's least-studied carnivores. Its biology and natural history remain poorly understood. It is thought to be tolerant of anthropogenic disturbance and inhabits a wide variety of habitat types, including primary and disturbed forests, open areas adjacent to forests, and oil palm plantations (*Md-Zain et al., 2019*). Within Borneo, this species is recorded frequently in the Malaysian state of Sabah, northern Borneo (*Samejima et al., 2016*; *Wong et al., 2017*), but much less frequently in South Kalimantan (*Higashide et al., 2018*) and exhibits a much localized distribution in northern Sarawak (*Giman & Jukie, 2012*). The driver of this patchy distribution across Borneo is presently unknown (*Samejima et al., 2016*; *Wong et al., 2017*). Understanding the Sunda stink-badger's genetic makeup, particularly through its mitochondrial genome, could provide insights into its evolutionary history and placement among the Mephitidae.

The mitochondrial genome is known for its fast evolution, maternal inheritance and simple structure, making it a reliable marker of molecular diversity (*Wang et al., 2022*). Mammalian mitochondrial genomes consist of closed circular strands containing a set of 13 protein-coding genes, two rRNA genes, and 22 tRNA genes, which have also been similarly reported in Mephitidae species (*Gibson et al., 2005*; *McDonough et al., 2022*). The genome is made up of two strands that are most commonly referred to as the heavy strand (H-strand) and the light strand (L-strand), distinguished by molecular weight differences caused by major differences in their base composition. Of the 13 protein-coding genes of mammalian mitochondrial genomes, 12 are located in the H-strand and only one is found in the L-strand. In addition, the genomes also contain areas known as the D-loop, which are believed to play functional roles in replication and transcription, as well as the origin of replication of the L-strand (OL), which is involved in DNA replication (*Gibson et al., 2005*).

Previous documentation indicates that access to mitochondrial genetic data from members of Mephitidae has been crucial for accurately defining species boundaries, identifying unrecognized species diversity within a geographic region, tracing complex evolutionary histories (such as secondary contacts between insular populations), understanding the influence of factors like climate change on phylogeographic diversification, and determining the timing and methods of specific geographic colonization. Such information plays a crucial role in guiding conservation decisions for the mephitids (*McDonough et al., 2022*; *Bolas et al., 2022*).

The absence of a complete mitochondrial genome sequence has been a drawback for *Mydaus*, as existing research entails limited choices of gene markers or the usage of partial genes (*Dragoo & Honeycutt, 1997*; *Md-Zain et al., 2019*). A complete mitochondrial genome for *Mydaus* will open up avenues for future investigations to ascertain true mitochondrial lineages. A profound understanding of mitochondrial sequence characterization can shed light on suitable regions to use as genetic markers. Additionally, a complete mitogenome may offer accurate signals for phylogenetic reconstruction compared to gene fragments (*Lan et al., 2024*). In this context, having a complete reference mitochondrial genome representing a genus can aid in the sequencing and assembly of additional species from the taxa.

Overall, the complete mitogenome sequence of *M. javanensis* will provide a wealth of information on its evolutionary history, genetic diversity, population dynamics, and relationships with other members of the Mephitidae. Thus, in the current study, we aimed to first determine and characterize the complete mitochondrial sequence of *M. javanensis*, and second, elucidate the taxonomic status and phylogenetic relationships between *M. javanensis* and other related taxa based on whole mitochondrial genomes. Additionally, we examined the *cyt b* gene, as it has the most available sequences in GenBank for comparison purposes within Mephitidae. This effort will not only add new information but also aid in conservation efforts and management strategies for this understudied member of the Mephitidae family.

## MATERIALS AND METHODS

### Sample collection and identification

Muscle tissues were collected from an adult male Sunda stink-badger (sample number MJ19) found as a road-killed animal during a routine field sampling trip in the Tawau region of Sabah, Malaysian Borneo (4°19′54.6″N 117°52′03.9″E). The specimen was placed in ice during field sampling and subsequently stored in a −20 °C freezer until further use. The sample was collected as part of a study on the ecology and conservation of Bornean carnivores, under an access license permit granted by the Sabah Biodiversity Centre (JKM/MBS.1000-2/2 JLD.12(48)) and under ethical review by the University of Oxford (Ref. No. APA/1/5/ZOO/NASPA/WildCRU/BorneanCarnivores).

### DNA extraction, PCR amplification and sequencing

DNA was isolated from approximately 25 mg of tissue using the DNeasy Blood and Tissue DNA Extraction Kit (Qiagen, Valencia, CA, USA) according to the manufacturer's

**Table 1  Details of the PCR primers and conditions used in the amplification of the Sunda stink-badger (*Mydaus javanensis*) mitogenome.**

| Primer name | | Sequence (5′–3′) | Amplification size | Annealing temperature (°C) |
|---|---|---|---|---|
| Leo16SLRpcr_F | Forward | CAGGACATCCCGATGGTGCAG | ~16 kb | 60 |
| Leo16SLRpcr_R | Reverse | ATCCAACATCGAGGTCGTAAAC | | |
| MJ19F | Forward | TGAAATTGACCTCCCCGTGA | 688 bp | 55 |
| MJ19R | Reverse | AGGCGCCTTTAGACTAACAGA | | |

protocol. After quality control using gel electrophoresis and UV spectrophotometry, the DNA sample was PCR-amplified with newly designed primers (Table 1) to capture the entire ~16 kb mitochondrial DNA, following a similar procedure developed by *Deiner et al. (2017)*. For this purpose, we used the high-fidelity PrimeSTAR® GXL polymerase (Takara Bio, Tokyo, Japan), which is specifically designed for long-range PCR amplification. The PCR was carried out in a total volume of 25 µL containing 30 ng of genomic DNA, 0.4 U of PrimeSTAR® GXL polymerase, 1x PrimerSTAR GXL buffer, 0.5 mm dNTPs (PrimerSTAR), and 10 pmol of primers Leo16SLRpcr_F and Leo16SLRpcr_R. The PCR amplification was performed as follows: pre-denaturation at 98 °C (10 s), followed by 40 cycles of denaturation at 98 °C (10 s), annealing at 60 °C (15 s) and extension at 68 °C (14 min). A final extension step at 68 °C for 5 min was included. The PCR product was then electrophoresed on a 0.8% agarose gel with 1x TBE buffer. The amplicon was excised from the gel and purified using the Wizard® SV Gel and PCR Clean-Up System (Promega, Madison, WI, USA). The sample was then sequenced using the Barcode Tagged Sequencing (BTSeq™) approach on an Illumina platform. Briefly, barcoded adapters are used to tag DNA fragments, which were then subsequently sequenced on an Illumina platform. Celemics' exclusive bioinformatics pipeline then organizes the sequencing reads based on molecular barcodes and aggregates them to rectify NGS errors, resulting in the generation of a complete DNA sequence. The sequencing service was provided by Celemics Inc (http://www.celemics.com).

In order to capture the regions flanking the primer binding sites of the circular mitogenome, we designed another set of primers (Table 1) to amplify a 688 bp region. This PCR was carried out as described above but with 15 pmol of primers MJ19F and MJ19R instead. The PCR amplification was as follows: pre-denaturation at 95 °C (5 min), followed by 40 cycles of denaturation at 95 °C (30 s), annealing at 55 °C (30 s) and extension at 72 °C (45 s). A final extension step at 72 °C for 10 min was included. The PCR product was electrophoresed on a 1.5% agarose gel, purified and subsequently sequenced bi-directionally using BigDye Terminator v3.1 on an ABI3130 Sequencer.

## Mitogenome annotation and analysis

The two high-quality reads from BTSeq and Sanger sequencing were manually assembled to form complete contiguous circular reads. The mitogenome was annotated using MITOS webserver (http://mitos2.bioinf.uni-leipzig.de; *Donath et al., 2019*). To ensure accuracy, the annotation, intergenic spacers, and overlapping regions between genes were manually

checked, counted and compared with complete and near complete mitogenomes of other related taxa from NCBI. All boundaries and secondary structures of tRNA gene were crossed-checked with tRNAscan-SE v2.0 (http://lowelab.ucsc.edu/tRNAscan-SE; *Chan & Lowe, 2019*) with the parameters: source = "Mito/Chloromast" and genetic code = "Vertebrate Mito and ARWEN v1.2 (*Laslett & Canbäck, 2008*), under default settings. A circular map of the mitogenome with all its respective features was drawn using the Proksee online tool (https://proksee.ca).

Base composition and relative synonymous codon usage (RSCU) were analyzed using MEGA-X (*Kumar et al., 2018*). Strand asymmetry was calculated using the formulas by *Perna & Kocher (1995)* *i.e.,* AT skew = $(A - T)/(A + T)$ and, GC skew = $(G - C)/(G + C)$.

## Characterization of the control region

The control region (CR) sequence of *M. javanensis* was mined from its complete mitogenome, which was sequenced in this study. The organization of *M. javanensis* was compared with those of other mephitids, whose CR regions were retrieved from their respective complete mitogenomes downloaded from NCBI GenBank. Incomplete sequences with gaps were discarded, prior to further analysis. The remaining sequences were examined for termination of the displacement loop (D-loop) motif, termination-associated sequences (TAS-A), and putative conserved sequence blocks, according to previous reports for *Lutra lutra* (Eurasian otter), *Conepatus chinga*, and *Conepatus leuconotus leuconotus* (*Ketmaier & Bernardini, 2005*). The alignment of these structural features was performed using MUSCLE in MEGA X, version 10.2.6 (*Kumar et al., 2018*). The repeats in the CR region among the available mephitids were identified using the Tandem Repeat Finder program (http://tandem.bu.edu/trf/trf.html) (*Benson, 1999*), under default settings.

## Phylogenetic analysis

In order to infer the phylogenetic relations of *M. javanensis* and other mephitids, concatenated nucleotide sequences were generated based upon 13 PCGs of the mitogenome. We also included members from Mustelidae, Procyonidae, Ailuridae and from the suborder Caniformia for comparison purposes. *Neofelis nebulosa* (Suborder: Feliformia) was treated as an outgroup. The sequences were downloaded from the GenBank online data repository. Complete mitogenomes were not available for *Conepatus humboldtii*, *Mephitis macroura* and *Mydaus marchei* and thus, excluded from the tree.

PartitionFinder 2.1.1 (*Lanfear et al., 2017*) was used to select the best substitution models and partition schemes (Data S1) with "greedy" algorithm and Bayesian information criterion (BIC), to be used in the subsequent phylogenetic analyses. Maximum likelihood (ML) and Bayesian inference (BI) approaches were applied for these analyses. The ML analysis was performed in IQ-TREE version 1.6.12 (*Nguyen et al., 2015*) with the node reliability assessed with 1,000 replicates of ultrafast likelihood bootstrap (*Minh, Nguyen & Von Haeseler, 2013*). The BI analysis was conducted with MrBayes on XSEDE v3.2.7a (*Ronquist et al., 2012*) available through the CIPRES Science Gateway (https://www.phylo.org/) (*Miller, Pfeiffer & Schwartz, 2010*). The Markov chain Monte Carlo (MCMC) runs were conducted for 10,000,000 generations and the trees were sampled every

1,000 generations with a burn-in of 25%. The software Tracer v1.7.2 (*Rambaut et al., 2018*) was employed to assess the parameters (effective sampling size for all parameters > 200).

In addition, we conducted phylogenetic analyses (ML and BI) using only *cyt b* gene sequences due to the scarcity of whole mitogenomes for the mephitids. Our analysis encompassed all accessible *cyt b* gene sequences from GenBank for members of the Mephitidae family. The first 37 nucleotide bases of the gene were trimmed for all species, considering the availability of partial gene for *Spilogale putorius putorius* and *Spilogale putorius ambarvalis*. The ML analysis was performed in IQ-tree version 1.6.12 (*Nguyen et al., 2015*) based on the best-substitution model (TIM2+F+I+G4) selected by ModelFinder (*Kalyaanamoorthy et al., 2017*) in the IQ-TREE package with 1,000 ultrafast bootstrap replicates (*Minh, Nguyen & Von Haeseler, 2013*). The BI analysis was executed with Mrbayes on XSEDE v3.2.7a (*Ronquist et al., 2012*) using similar parameters as those employed for the BI analysis conducted for the concatenated sequences of 13 PCGs. The best-fit substitution model (TIM2+I+G) was determined *via* Jmodeltest2 (*Darriba et al., 2012*) and these were available through CIPRES Science Gateway (https://www.phylo.org/) (*Miller, Pfeiffer & Schwartz, 2010*).

## RESULTS

### Mitogenome organization and composition

The complete mitogenome of *M. javanensis* sample MJ19 was 16,391 bp and the GenBank accession number is OP442081. It was sequenced to a high quality at about $1100\times$ coverage. The size of the mitogenome was within the range of the complete mitogenomes from other mephitids (Table 2). It comprises of the typical 13 protein-coding, 22 tRNAs, two ribosomal RNA genes, one control region (CR) and an L-strand replication origin ($O_L$). The *ND6* gene, along with eight other tRNAs (*trnQ, trnA, trnN, trnC, trnY, trnS2, trnE* and *trnP*), was positioned in the reverse direction (Fig. 1). All other genes and miscellaneous regions were in the forward direction. The *trnL* and *trnS* were made up of two major codons *i.e., trnL1 (UUR)* and *trnL2 (CUN)* and *trnS1 (AGY)* and *trnS2 (UCN)*, respectively. A total of nine pairs of genes (or regions) overlapped with one another, with overlaps ranging from 1 to 43 nucleotides (Table 3). We noticed that five PCGs had incomplete stop codons and likely rely on post-transcriptional polyadenylation (TAA) for termination.

With regards to the base composition, the mitogenome was skewed with a clear bias towards an A+T content of 61.9%, while G+C was 38.1% (Fig. 2, Data S2). Composition analysis revealed that the mitogenome exhibited a positive AT (0.095) and a negative GC skew (−0.307) as a whole, as well as in the 13 PCGs (AT skew: 0.045; GC skew: −0.337), 2 rRNAs (AT skew: 0.201; GC skew: −0.099) and the control region (AT skew: 0.039; GC skew −0.253). However, for the tRNA genes, both the AT and GC skews were positive, at 0.031 and 0.073, respectively (Fig. 2, Data S2).

### Protein coding genes and codon usage

The mitogenome was comprised of the typical 13 protein coding genes (PCGs) found in mammals. The concatenated lengths of the 13 PCGs were 11,424 bp and a total of 3,808 codons were involved in protein translation. Based on the codon count, leucine was the

**Table 2  Availability of mitogenome and *cyt b* gene sequences for the 12 extant species in the family Mephitidae.** No mitogenome sequences were available for *C. humboldtii*, *M. macroura*, *M. javanensis* and *M. marchei*. Of these four, *cyt b* gene sequences were only available for *M. macroura* and *M. javanensis*.

| Genera | Species | Availability of mitogenome sequences in GenBank | Availability of *cytb* gene sequences in GenBank |
|---|---|---|---|
| Conepatus | *C. chinga* | NC_042596 (complete) | Available |
| | *C. humboldtii* | None | None |
| | *C. leuconotus* | MW205848 (partial genome) | Available |
| | *C. semistriatus* | MW205849 (partial genome) | Available |
| Mephitis | *M. macroura* | None | KY026063 |
| | *M. mephitis* | NC_020648 (complete) | Available |
| Mydaus | *M. javanensis* | None | AB564095 |
| | *M. marchei* | None | None |
| Spilogale | *S. angustifrons* | MW205870 (complete) | Available |
| | | MW205885 *S. angustifrons yucatanensis* (complete) | Available |
| | *S. gracilis* | MW205896 *S. gracilis gracilis* (complete) | Available |
| | | MW205880 *S. gracilis leucoparia* (complete) | Available |
| | | MW205868 *S. gracilis martirensis* (complete) | Available |
| | | MW205862 *S. gracilis lucasana* (complete) | Available |
| | *S. putorius* | NC_010497 (complete) | Available |
| | | MW205890 *S. putorius interrupta* (complete) | Available |
| | | *S. putorius putorius* (None) | MG753651 |
| | | *S. putorius ambarvalis* (None) | MG753655 |
| | *S. pygmaea* | MW205863 (partial) | Available |

most common amino acid (589), followed by threonine (332) and isoleucine (325). It was interesting to note that the RSCU indicated that degenerate codons were biased towards using more A and C at the third codon compared to G and U (Fig. 3).

## Transfer RNA and ribosomal RNA genes

The *M. javanensis* mitogenome contained the typical 22 tRNA genes and their lengths ranged from 59 (*trnS1*) to 75 (*trnL2*). All tRNA exhibited the typical cloverleaf secondary structure, with the exception of *trnS1* $^{AGY}$ which lacks the dihydrouridine loop (Fig. 4). There was a total of 38 mismatches (U-G, U-U, C-A, A-A, A-G and C-U) with the U-G (74%) being the most common.

In addition, the two rRNA genes (*rrnL* and *rrnS*) were highly conserved across the mephitids. The putative lengths of *rrnL* and *rrnS* were 1,572 and 957 bp, respectively, and both had a positive AT skew and a negative GC skew (Table 3, Fig. 2 and Data S2).

## Control Region (CR)

At position 162 of the *M. javanensis*'s control region, we identified a motif homologous to the termination-associated sequences (TAS-A). The motif of the D-loop termination of the CR (GCCCC) was identified few nucleotides upstream of TAS-A. In addition, all the eight putative conserved sequence blocks (CSB1-3 and B-F) were identified within the CR region. A single region with tandem repeats, referred to as RS3, was discovered in between
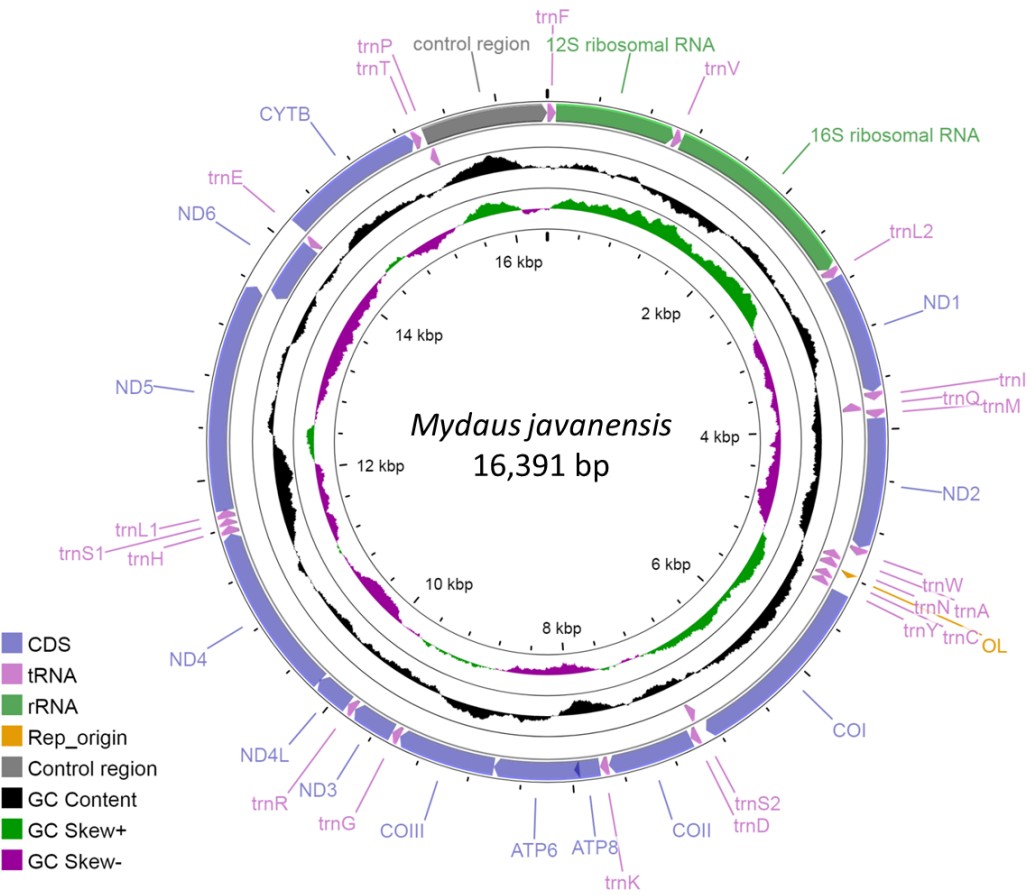

**Figure 1** **Circular map of the Sunda stink-badger (*Mydaus javanensis*) mitochondrial genome.** The protein coding and ribosomal genes are shown in standard abbreviations. Different colors represent the different gene blocks or regions in the mitogenome.

CSB-1 and CSB-2. A schematic diagram illustrating the organization of the control region of *M. javanensis* is shown in Fig. 5.

A comparative analysis of the control region was conducted according to previous reports on Eurasian otter and other mephitids (Fig. 6). The analysis reveals the presence of fundamental structures, such as the D-loop termination, TAS-A, and multiple conserved sequence blocks, in all the mephitids. The identification and comparison of the tandem repeats among mephitids were also conducted, revealing differences in the distribution of the repeat regions (Table 4). Similar to *M. javanensis*, mephitids such as *Mephitis mephitis*, *Conepatus chinga*, and *Conepatus leuconotus leuconotus* display a single repeat region located in the right domain, between CSB-1 and CSB-2. Conversely, *Spilogale angustifrons* showed the presence of a repeat region in the left domain. In contrast, *Spilogale putorius* exhibited two repeat regions, one in the right domain and the other in the left domain. Interestingly, the Tandem Repeat Finder did not detect any repeats in either of the subspecies of *Spilogale gracilis*. Both *Spilogale gracilis gracilis* and *Spilogale gracilis leucoparia* had the smallest control region sizes, with 819 bp and 853 bp, respectively

**Table 3  Composition and annotation of the newly sequenced mitogenome of the Sunda stink-badger (*Mydaus javanensis*).**

| | Feature name | Type | Start position | End position | Size | Direction | Start codon | Stop codon | Intergenic nucleotides |
|---|---|---|---|---|---|---|---|---|---|
| 1 | *trnF* | tRNA | 1 | 69 | 69 | Forward | – | – | 0 |
| 2 | *rrnS* | rRNA | 70 | 1,026 | 957 | Forward | – | – | 0 |
| 3 | *trnV* | tRNA | 1,027 | 1,094 | 68 | Forward | – | – | 0 |
| 4 | *rrnL* | rRNA | 1,095 | 2,666 | 1,572 | Forward | – | – | 0 |
| 5 | *trnL2* | tRNA | 2,667 | 2,741 | 75 | Forward | – | – | 3 |
| 6 | *ND1* | gene | 2,745 | 3,700 | 956 | Forward | ATG | TA(A) | 0 |
| 7 | *trnI* | tRNA | 3,701 | 3,769 | 69 | Forward | – | – | −3 |
| 8 | *trnQ* | tRNA | 3,767 | 3,839 | 73 | Reverse | – | – | 1 |
| 9 | *trnM* | tRNA | 3,841 | 3,910 | 70 | Forward | | | 0 |
| 10 | *ND2* | gene | 3,911 | 4,952 | 1,042 | Forward | ATA | T(AA) | 0 |
| 11 | *trnW* | tRNA | 4,953 | 5,019 | 67 | Forward | – | – | 13 |
| 12 | *trnA* | tRNA | 5,033 | 5,100 | 68 | Reverse | – | – | 1 |
| 13 | *trnN* | tRNA | 5,102 | 5,174 | 73 | Reverse | – | – | 0 |
| 14 | O$_L$ | rep_origin | 5,175 | 5,207 | 33 | Forward | – | – | −1 |
| 15 | *trnC* | tRNA | 5,207 | 5,272 | 66 | Reverse | – | – | 0 |
| 16 | *trnY* | tRNA | 5,273 | 5,339 | 67 | Reverse | – | – | 1 |
| 17 | *COI* | gene | 5,341 | 6,885 | 1,545 | Forward | ATG | TAA | −3 |
| 18 | *trnS2* | tRNA | 6,883 | 6,951 | 69 | Reverse | – | – | 5 |
| 19 | *trnD* | tRNA | 6,957 | 7,023 | 67 | Forward | – | – | 0 |
| 20 | *COII* | gene | 7,024 | 7,707 | 684 | Forward | ATG | TAA | 3 |
| 21 | *trnK* | tRNA | 7,711 | 7,779 | 69 | Forward | – | – | 1 |
| 22 | *ATP8* | gene | 7,781 | 7,984 | 204 | Forward | ATG | TAA | −43 |
| 23 | *ATP6* | gene | 7,942 | 8,622 | 681 | Forward | ATG | TAA | −1 |
| 24 | *COIII* | gene | 8,622 | 9,405 | 784 | Forward | ATG | T(AA) | 0 |
| 25 | *trnG* | tRNA | 9,406 | 9,474 | 69 | Forward | – | – | 0 |
| 26 | *ND3* | gene | 9,475 | 9,821 | 347 | Forward | ATA | TA(A) | 0 |
| 27 | *trnR* | tRNA | 9,822 | 9,889 | 68 | Forward | – | – | 0 |
| 28 | *ND4L* | gene | 9,890 | 10,186 | 297 | Forward | ATG | TAA | −7 |
| 29 | *ND4* | gene | 10,180 | 11,557 | 1,378 | Forward | ATG | T(AA) | 0 |
| 30 | *trnH* | tRNA | 11,558 | 11,626 | 69 | Forward | – | – | 0 |
| 31 | *trnS1* | tRNA | 11,627 | 11,685 | 59 | Forward | – | – | 0 |
| 32 | *trnL1* | tRNA | 11,686 | 11,755 | 70 | Forward | – | – | −9 |
| 33 | *ND5* | gene | 11,756 | 13,576 | 1,830 | Forward | ATA | TAA | −17 |
| 34 | *ND6* | gene | 13,560 | 14,087 | 528 | Reverse | ATG | TAA | 0 |
| 35 | *trnE* | tRNA | 14,088 | 14,156 | 69 | Reverse | – | – | 4 |
| 36 | *CYTB* | gene | 14,161 | 15,300 | 1,140 | Forward | ATG | AGA | 0 |
| 37 | *trnT* | tRNA | 15,301 | 15,369 | 69 | Forward | – | – | −1 |
| 38 | *trnP* | tRNA | 15,369 | 15,434 | 66 | Reverse | – | – | −40 |
| 39 | Control region | D-loop | 15,395 | 16,391 | 997 | Forward | – | – | 0 |

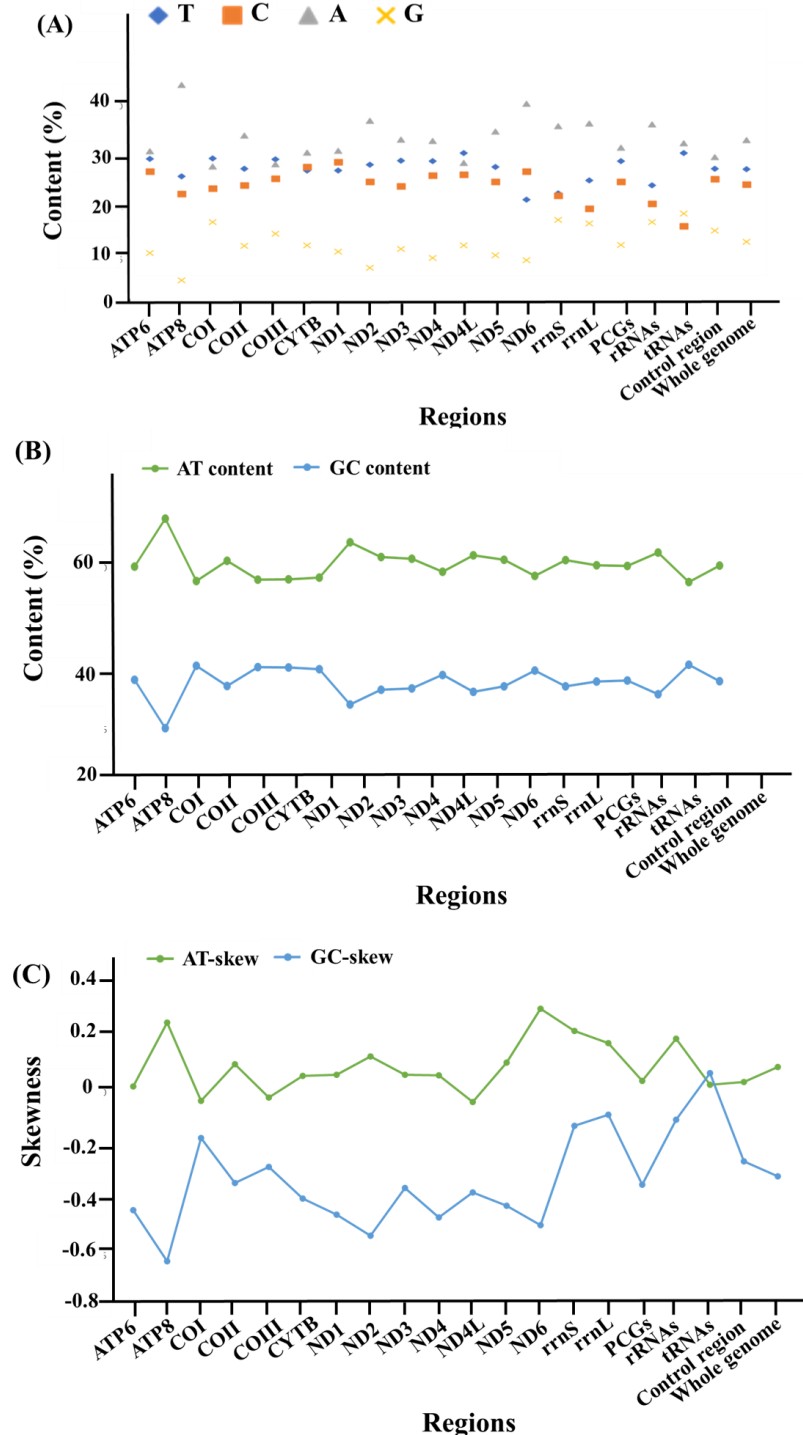

**Figure 2  The base composition and skewness of the complete mitogenome of *M. javanensis*.** (A–B) Base composition, (C) AT- and GC- skews.

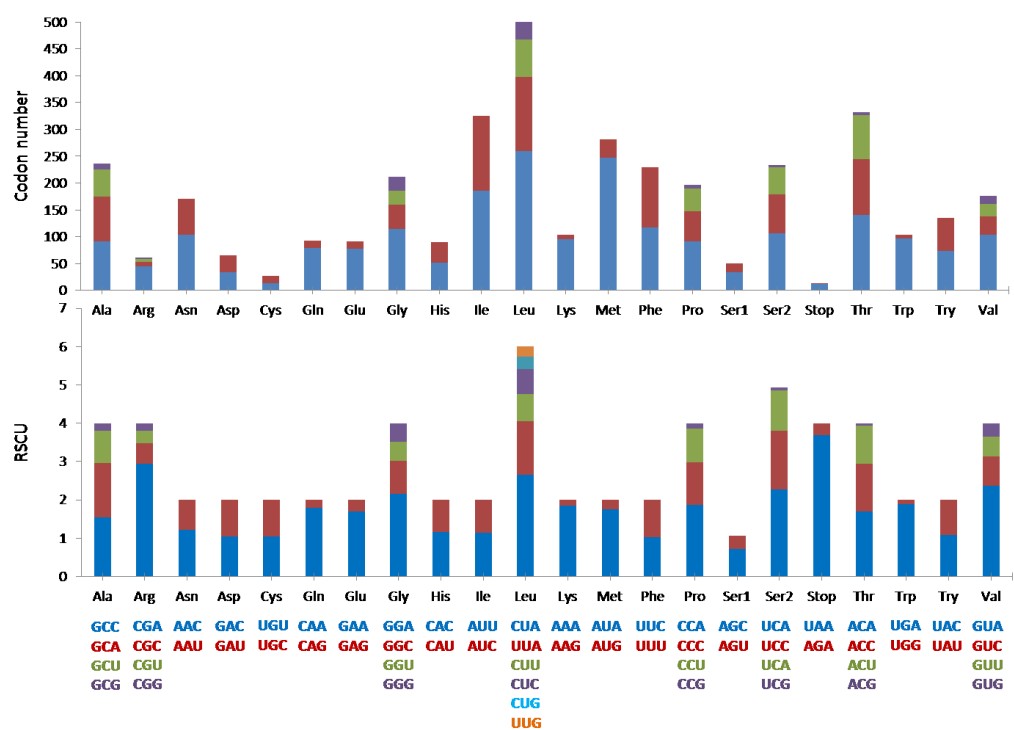

**Figure 3** The codon number and relative synonymous codon usage (RSCU) of the 13 protein coding genes (PCGs) in *Mydaus javanensis* mitogenome.

(Table 4). Overall, the mitochondrial control region maintains conservation in its basic structures, with differences appearing in the tandem repeat regions.

## Phylogenetic analysis

Phylogenetic trees of ML (Fig. 7) and BI (Fig. 8) analyses were built on concatenated sequences of 13 PCGs from 30 species, with 13 members representing the Mephitidae. Overall, both the trees were highly congruent and received high bootstrap support for the majority of branches, accurately clustering the species within its specific genera and family. Both the topologies confirmed the monophyly of the family Mephitidae which included the members representing *Spilogales*, *Mephitis*, *Conepatus* and *Mydaus*. *M. javanensis* MJ19 (Borneo, Malaysia) was placed into this monophyletic group together with the other member from Java, Indonesia, with high nodal supports from both the trees (boostrap support values (BS) = 100 and Bayesian posterior probability (BPP) was equal to 1.00). Meanwhile, we have also observed that Mustelidae forms a sister clade with Procyonidae.

Meanwhile, ML (Fig. 9) and BI (Fig. 10) trees constructed based on *cyt b* gene included additional species within Mephitidae. The resulting trees generated the same topology and BI analysis provided increased resolution with stronger support than ML analysis. The trees correctly grouped the Javanese and Bornean Sunda stink-badgers with high bootstrap values (BS = 100). However, we noticed 27 nucleotide variations between the two individuals in the partial *cyt b* gene (the first 37 nucleotide bases were trimmed to

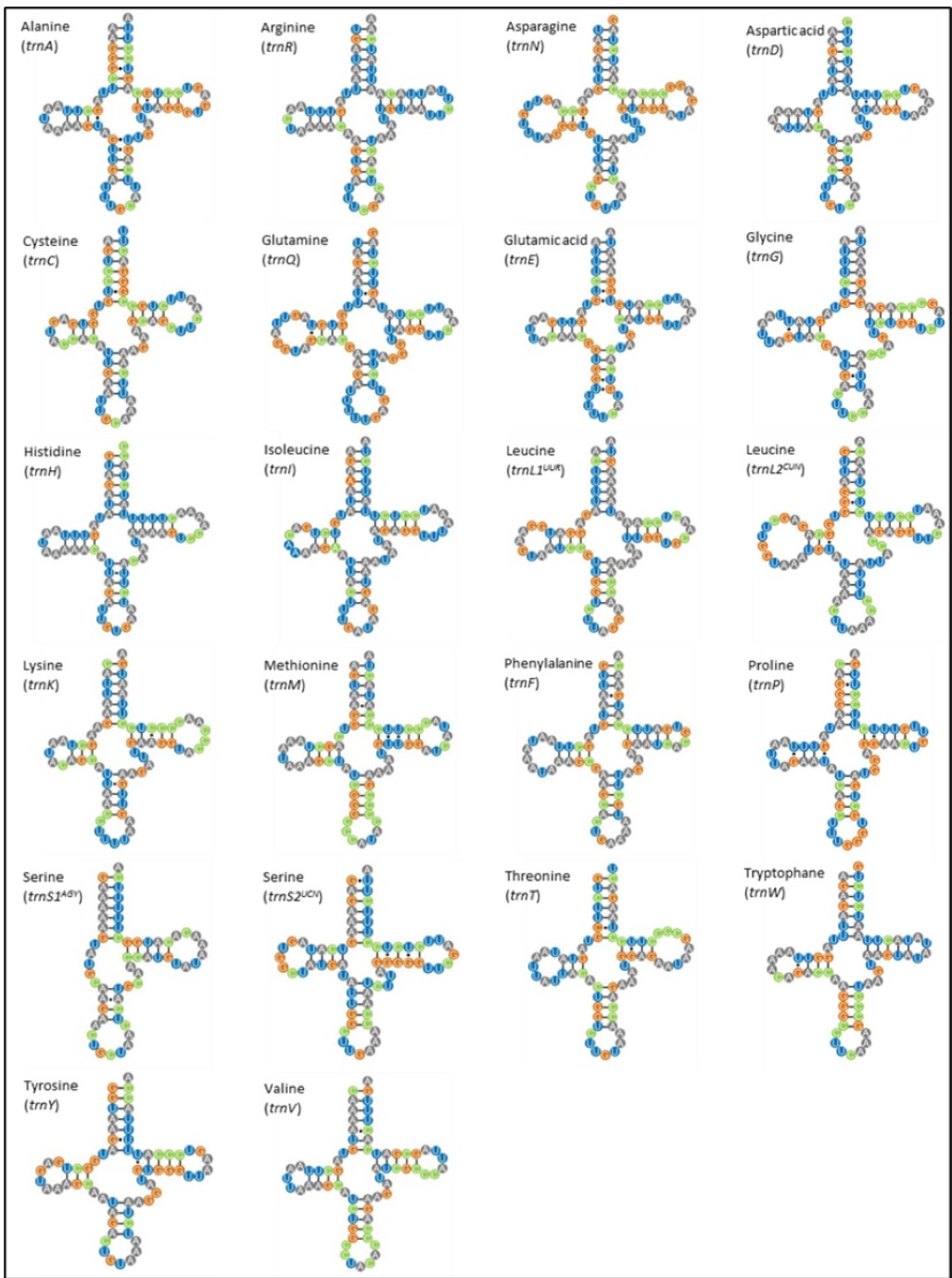

**Figure 4** **Putative secondary structures of *Mydaus javanensis* tRNAs indicating the typical clover leaf shape consisting of the acceptor stem, DHU loop, anticodon loop, T Ψ C Loop, variable arm and the discriminator base.** The tRNAs are labeled with their corresponding amino acids. The dashes (-) indicate the Watson-Crick bonds and the dot (∗) indicates a mispairing between the nucleotides.

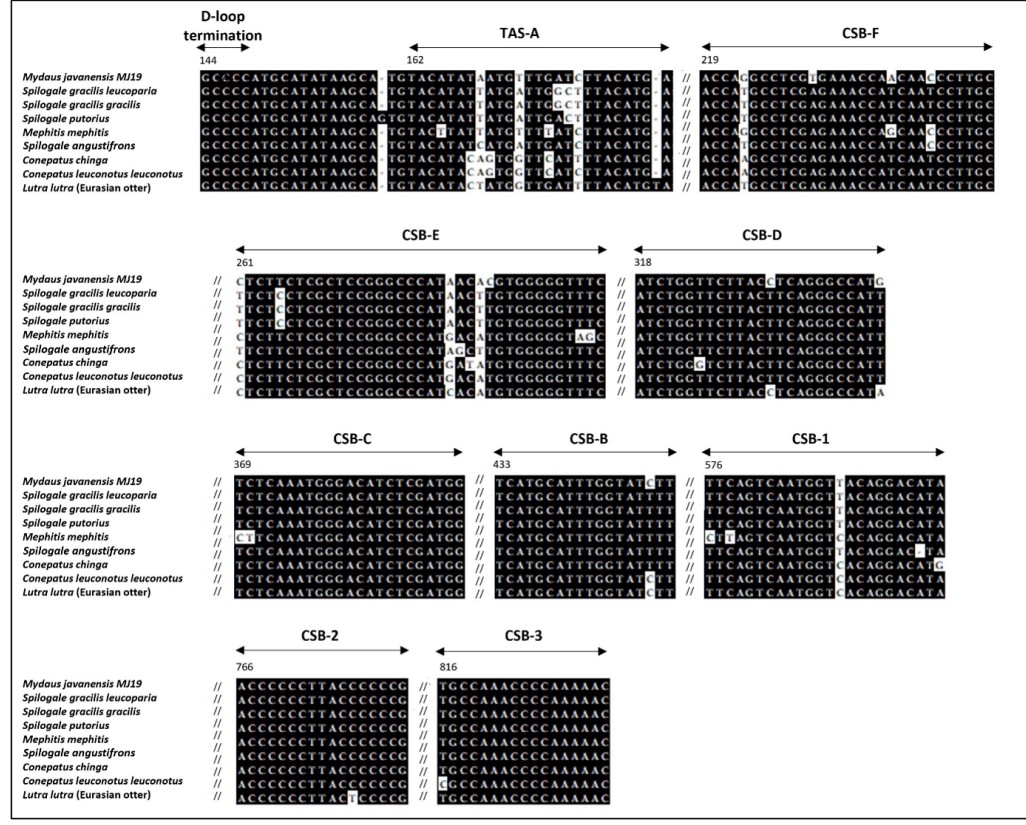

**Figure 5** Schematic diagram of the organization of the mitochondrial control region (CR) of *Mydaus javanensis.* The CR is shown to be bound by the tRNA$_{Pro}$ and tRNA$_{Phe}$ genes. The termination of the displacement loop (D-loop) motif (D-loop termination; blue block), termination-associated sequence (TAS-A; orange block) and conserved sequence blocks (CSBs; green blocks) were specified according to *Ketmaier & Bernardini (2005)*. The grey striped region indicates that the repeat region, RS3 (position: 607-727) is located between CSB1 and CSB2.

**Figure 6** Alignment of the D-loop termination, TAS-A, and eight CSBs of mitochondrial control region (CR) of *Mydaus javanensis* and other related species included in the study. The sequences of *Conepatus chinga*, *Conepatus leuconotus leuconotus* and *Lutra lutra* serve as references for the alignments, according to *Ketmaier & Bernardini (2005)*. The black background indicates conserved bases while white background indicates variations observed in the nucleotides. Numbers above the alignment point out the first nucleotide positions of each region based on mitochondrial control region of *Mydaus javanensis*.

**Table 4  Details of tandem repeat regions detected within mitochondrial control regions of mephitids.**

| Species (Genbank accession number) | Gene size (bp) | No. of tandem repeat regions | Tandem repeat positions in control region |
|---|---|---|---|
| *Mydaus javanensis* MJ19 (OP442081) | 997 | 1 | a. Right domain (Between CSB-1 & CSB-2) |
| *Spilogale gracilis leucoparia* (MW205880) | 853 | None | None |
| *Spilogale gracilis gracilis* (MW205896) | 819 | None | None |
| *Spilogale putorius* (NC_010497) | 1,138 | 2 | a. Right domain (Between CSB-1 & CSB-2) b. Left Domain |
| *Mephitis mephitis* (NC_020648) | 1,103 | 1 | a. Right domain (Between CSB-1 & CSB-2) |
| *Spilogale augustifrons* (MW205870) | 899 | 1 | a. Left domain |
| *Conepatus chinga* * (NC_042596) | 1,067 | 1 | a. Right domain (Between CSB-1 & CSB-2) |
| *Conepatus leuconotus leuconotus* * (AY159816) | 1,218 | 1 | a. Right domain (Between CSB-1 & CSB-2) |

**Notes.**
The asterisk (*) denotes that the sequence is treated as a reference, according to *Ketmaier & Bernardini (2005)*.

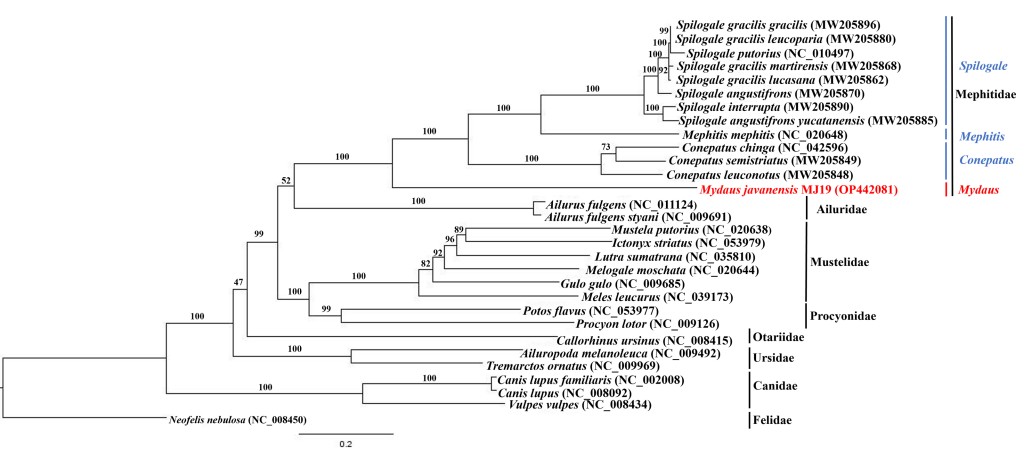

**Figure 7  Maximum likelihood (ML) phylogenetic placement of the Sunda stink-badger (*Mydaus javanensis*) within the Mephitidae family and other selected members of suborder Caniformia.** The tree was based on the concatenated sequences of 13 protein coding genes (PCGs). The clouded leopard (*Neofelis nebulosa*) was used as an outgroup. The sequence characterized in this study is highlighted in red. Genera of the mephitids are indicated in blue highlights. The bootstrap values of the branches were displayed at each node.

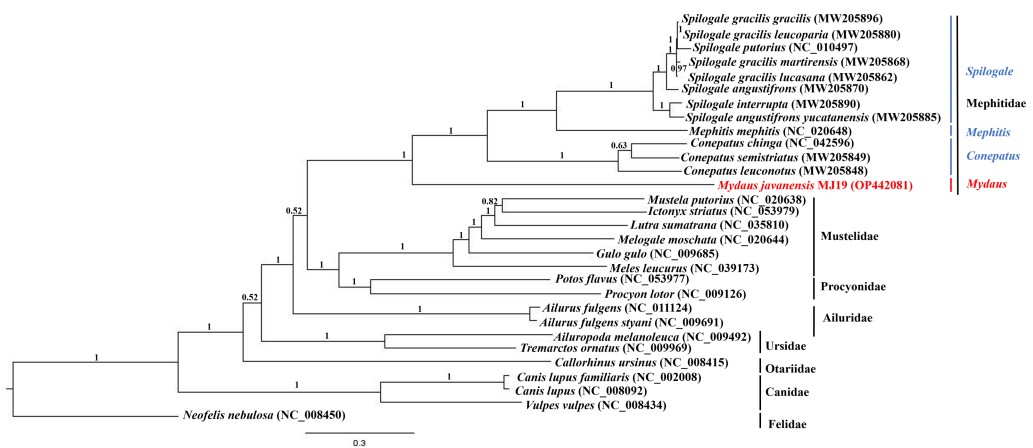

**Figure 8** Bayesian inference (BI) phylogenetic placement of the Sunda stink-badger (*Mydaus javanensis*) within the Mephitidae family and other selected members of suborder Caniformia. The tree was based on the concatenated sequences of 13 protein coding genes (PCGs). The clouded leopard (*Neofelis nebulosa*) was used as an outgroup. The sequence characterized in this study is highlighted in red. Genera of the mephitids are indicated in blue highlights. The probability values of the branches were displayed at each node.

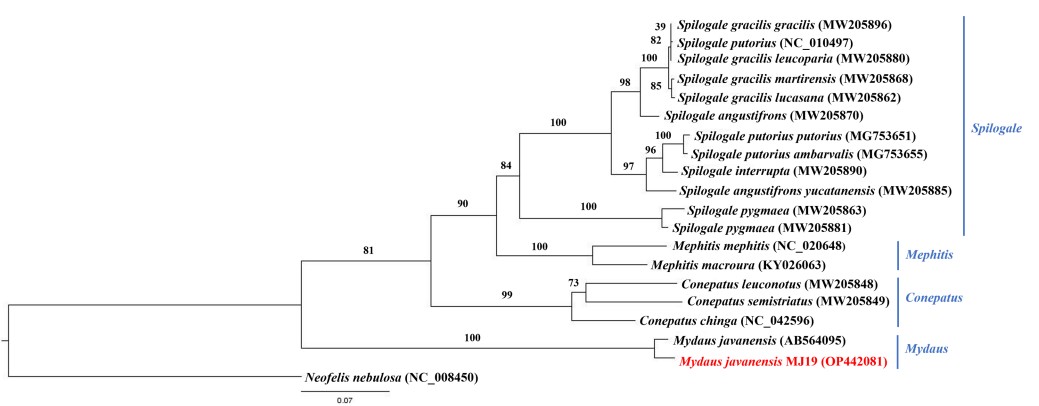

**Figure 9** Maximum likelihood (ML) phylogenetic placement of the Sunda stink-badger (*Mydaus javanensis*) within the Mephitidae family. The tree was based on the *cyt b* gene. The clouded leopard (*Neofelis nebulosa*) was used as an outgroup. The sequence characterized in this study is highlighted in red. Genera of the mephitids are indicated in blue highlights. The bootstrap values of the branches were displayed at each node.

standardize the lengths of all sequences under study) (Data S3 and Data S4). All nucleotide variants at the third codon position were degenerate and did not result in changes in the amino acid. However, when the nucleotide variants were in the first codon base, it invariably led to seven non-synonymous amino acid changes (Data S5).

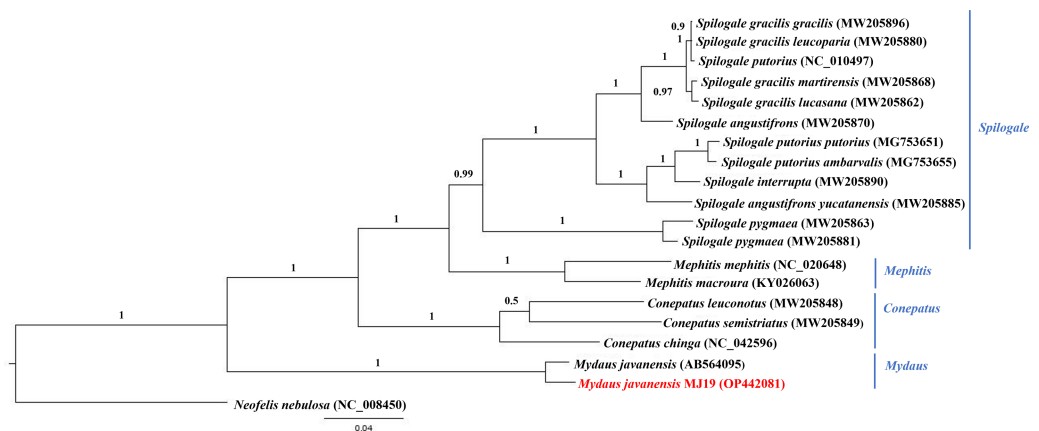

**Figure 10  Bayesian inference (BI) phylogenetic placement of the Sunda stink-badger (*Mydaus javanensis*) within the Mephitidae family.**  The tree was based on the *cyt b* gene. The clouded leopard (*Neofelis nebulosa*) was used as an outgroup. The sequence characterized in this study is highlighted in red. Genera of the mephitids are indicated in blue highlights. The probability values of the branches were displayed at each node.

## DISCUSSION

The mitogenome of the Sunda stink-skunk was sequenced and characterized. This is the first publicly available complete mitogenome for the species. The information provided here has added new information to the relatively understudied members of Mephitidae family. It is important to note that we still do not have mitogenomes for another three mephitids (*Conepatus humboldtii*, *Mephitis macroura* and *Mydaus marchei*), while some mitogenomes (*C. leuconotus*, *C. semistriatus*, *S. pygmaea*) were incomplete and contained stretches of Ns in their sequences (Table 2). The primers used in the study managed to amplify the entire mitochondrial DNA in a single PCR following the approach originally used by *Deiner et al. (2017)* and *Kato-Unoki, Umemura & Tashiro (2020)*. Targeting the entire mitogenome in a single PCR is preferable as it prevents the amplification of nuclear encoded pseudo mitochondrial genes and avoids the misalignment of gene order during assembly and annotation (*Parr et al., 2006*; *Montaña Lozano et al., 2022*). However, one drawback of this approach is that DNA sequencing is unable to capture sequences at the primer binding sites, hence, a second set of primers were needed to target this region. Our primers were based on conserved regions and may be useful to other researchers who are working on mephitids or other closely related species. In addition, these primers could be used to fill up the gaps in the partial mitogenomes mentioned in Table 2.

The structure of the *Mydaus javanensis* MJ19 mitogenome was similar to other vertebrates with the typical 13 protein-coding genes, 22 tRNAs, two ribosomal RNA genes, one control region (CR) and an L-strand replication origin ($O_L$) (*Boore, 1999*; *Pereira, 2000*; *Montaña Lozano et al., 2022*). These components make up the mammalian mitochondrial oxidative phosphorylation (OXPHOS) system (*Signes & Fernandez-Vizarra, 2018*; *Shokolenko & Alexeyev, 2022*). Overlapping among 10 gene pairs was observed and

such occurrence has been proposed to extend genetic information within the constraints of limited genome size (*Sun et al., 2020*).

The AT and GC skew of *M. javanensis* was similar to other members of the Mephitidae and to that reported previously in Mustelidae (*Skorupski, 2022*). It is interesting to note that the bias toward A and T and against G and C is a common feature in metazoan mitogenomes. This naturally leads to a subsequent bias in the corresponding encoded amino acids as seen in the codon usage. However, when compared to members of Proconidae, Ailuridae, Canidae and Ursidae, we noticed a smaller GC skew (an average of −0.268) in the species (*Skorupski, 2022*), indicating a smaller proportion of G bases compared to Cs. This is reflected in *M. javanensis*, where the degenerate codons were biased towards A and C nucleotides at the third codon position.

With regards to the 22 tRNA genes in *M. javanesis*, we observed a typical and conserved arrangement as found in most vertebrates (*Watanabe, Suematsu & Ohtsuki, 2014*). One unique feature is the lack of the dihydrouridine (DHU) loop in $trnS1^{AGY}$ which is commonly observed in all vertebrates (*Pereira, 2000*). The DHU loop is important as it is involved in the aminoacylation of the tRNA molecule. The loop functions as a recognition site for aminoacyl-tRNA synthetase (*Watanabe, Suematsu & Ohtsuki, 2014*). In contrast, $trnS2^{UCN}$ has the complete typical cloverleaf pattern features of tRNAs including the DHU loop. While both the $trnS1^{AGY}$ and $trnS2^{UCN}$ are capable of translation on the ribosome, *Hanada et al. (2001)* has shown that the lack of a DHU loop in $trnS1^{AGY}$ results in considerably lower translational activity. This explains why $trnS2^{UCN}$ is the preferred tRNA for Serine as observed by the codon count and RSCU distribution in *M. javanensis* (Fig. 3).

With regards to the analysis of the control region (CR) organization in *M. javanensis*, we discovered the presence of the fundamental conserved structures similar to those previously identified in the Eurasian otter (a mustelid) as well as other mephitids, such as the D-loop termination motif, TAS-A and eight conserved sequence blocks (*Ketmaier & Bernardini, 2005*). Additionally, the CR of mammals is known to have two potential locations with tandem repeats: one in the left and the other in the right domains of the gene, respectively (*Wilkinson et al., 1997*). In *M. javanensis* we found a single repeat region (RS3) which is placed between CSB-1 and CSB-2, similar to various other skunks (*Ketmaier & Bernardini, 2005*). Notably, this location has only been described in mammals (*Ketmaier & Bernardini, 2005*). As such, CSB-1 and CSB-2 can serve as potential primer binding sites for future amplifications of RS3 regions, especially for the species under *Mydaus*.

Additionally, a comparative analysis of CR structural organization among mephitids showed that the CR region is well-structured, with the central region being highly conserved and tandemly repeated sequences occurring only within the two peripheral domains. These peripheral domains are rapidly evolving regions characterized by a high rate of nucleotide substitutions and variations in the copy number of tandem repeats. This variability in the number of tandemly repeated sequences is considered as the pivotal source of mitochondrial DNA length variation in animals (*Brown et al., 1986*). Consistent with this, we found that species lacking the repeat regions had the smallest overall control regions sizes (Table 4). Species within the Mephitidae family exhibited diverse distributions of repeat regions. Some had none, some had two, and those with a single repeat region had it located in

either the right or left domains. Interestingly, the distribution of repeat regions was found to be species-specific. In future studies, additional species within the genus *Mydaus* can be investigated to gain further insights into the distribution and evolutionary patterns of tandem repeats in the control region (CR).

The ML and BI trees based on the concatenated sequences of 13 Protein Coding Genes (PCGs) correctly grouped the nine (out of 12) extant species of mephitids in four 4 genera of the family Mephitidae. *M. javanensis* was accurately grouped with the mephitids and not with the mustelids. The genus *Mydaus* was initially placed as a member of Mustelidae until DNA studies revealed that the skunks, along with stink-badgers (*Mydaus* spp.), belong to a separate family (Mephitidae), which is highly divergent from the mustelids (*Koepfli, Dragoo & Wang, 2017*). The phylogeny using the complete set of core protein coding genes of the mitochondrial genome conducted in this study, is in concordance with this finding. Additionally, both the ML and BI trees reflected a common observance of Mustelidae forming sister clade with Procyonidae. The sister-group relationship between these two families is congruent to previous documentations, supported by morphological characters and molecular data (*Sato et al., 2003*).

To utilize the sequences available in GenBank for the mephitids, we used the *cyt b* gene sequences to perform ML and BI analyses. The sequence variability of *cyt b* gene makes it a popular choice in systematic studies of numerous organisms to resolve divergences at various taxonomic levels (*Castresana, 2001*). The efficient utilization of the gene fragment has previously been reported in the genetic studies of stink-badgers and Eurasian badgers (*Dragoo & Honeycutt, 1997*; *Md-Zain et al., 2019*). In the current study, the ML and BI analyses of *M. javanensis* MJ19 has been conducted with currently available species and subspecies of Mephitidae. In both topologies, sample *M. javanesis* MJ19, from northern Borneo (Sabah, Malaysia) was grouped together with the Javanese sample from Indonesia with high support values (BS = 100 and BPP = 1.0). However, there were 27 nucleotide variations between the two individuals in the corresponding *cyt b* gene analysed, among which seven variants found to be non-synonymous. These results indicate the interspecies variations that can differentiate the Sunda stink-badgers from Java and Borneo. It would be worthwhile to sequence the complete mitogenomes from the Javanese species and surrounding Southeast Asian populations to determine its genetic diversity, enhancing our understanding of the evolutionary patterns of *Mydaus*.

## CONCLUSION

In the present study, we sequenced the first mitogenome of the Sunda stink-skunk (*M. javanensis*), revealing key insights into its genomic structures and characteristics. We found that the genome size of *M. javanensis* (16,391 bp) is consistent with other metazoan mitogenomes, and it shows the usual A+T content bias prevalent in these genomes. Our analyses shed new light on the organization of the mitochondrial control region (CR) in *M. javanensis* compared to other mephitids, highlighting species-specific variations in the presence and distribution of the tandem repeats. The phylogenetic relationships inferred from both ML and BI methods, based on concatenated sequences of 13 PCGs and the *cyt*

*b* gene, consistently support the monophyly of Mephitidae. Notably, *M. javanensis* clusters with *Conepatus*, *Mephitis*, and *Spilogale*, reinforcing its taxonomic position among extant Mephitidae species. These findings presented here contribute valuable information to the relatively understudied members of the Mephitidae family.

The future direction of the study should include more samples of *M. javanensis* from surrounding Southeast Asian populations, as well as from closely related species, especially *Mydaus marchei*, being a sister taxon to *M. javanensis*, to gain a more comprehensive understanding of their genetic diversity, evolutionary history, and potential interspecies gene flow. This effort would ultimately support conservation initiatives and inform management strategies for these species.

### Funding
This study was supported by the Robertson Foundation, Dr. Holly Reed Conservation Fund, Point Defiance Zoo & Aquarium and Universiti Malaysia Sabah (No. LPA2006). The funders had no role in study design, data collection and analysis, decision to publish, or preparation of the manuscript.

### Grant Disclosures
The following grant information was disclosed by the authors:
The Robertson Foundation, Dr. Holly Reed Conservation Fund, Point Defiance Zoo & Aquarium and Universiti Malaysia Sabah: No. LPA2006.

### Competing Interests
The authors declare there are no competing interests.

### Author Contributions
- Vijay Kumar Subbiah conceived and designed the experiments, performed the experiments, analyzed the data, prepared figures and/or tables, authored or reviewed drafts of the article, and approved the final draft.
- Chrishen Robert Gomez performed the experiments, analyzed the data, authored or reviewed drafts of the article, and approved the final draft.
- Dexter Miller Robben performed the experiments, analyzed the data, prepared figures and/or tables, and approved the final draft.
- Ranjita Subramaniam analyzed the data, prepared figures and/or tables, authored or reviewed drafts of the article, and approved the final draft.
- Andrew James Hearn conceived and designed the experiments, performed the experiments, analyzed the data, authored or reviewed drafts of the article, and approved the final draft.

### Animal Ethics
The following information was supplied relating to ethical approvals (i.e., approving body and any reference numbers):

The sample was collected as part of a study of the ecology and conservation of Bornean carnivores, under an access license permit from Sabah Biodiversity Centre (JKM/MBS.1000-2/2 JLD.12(48)) and under ethical review from the University of Oxford (Ref. No. APA/1/5/ZOO/NASPA/WildCRU/BorneanCarnivores).

### Field Study Permissions

The following information was supplied relating to field study approvals (i.e., approving body and any reference numbers):

Field experiments were approved by the Sabah Biodiversity Centre (JKM/MBS.1000-2/2 JLD.12(48)).

### Data Availability

The genome sequence data that support the findings of this study are available at GenBank: OP442081.

### Supplemental Information

Supplemental information for this article can be found online at http://dx.doi.org/10.7717/peerj.18190#supplemental-information.

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
