# Peer review of "Characterization of the complete mitochondrial genome of the Sunda stink-badger (Mydaus javanensis) from the island of Borneo"

_PeerJ, doi:10.7717/peerj.18190_

## Round 0.1 · original submission · Major Revisions

I regret to report that the manuscript is unfit for publication in its present form.
I agree with the Reviewers’ criticisms, and I particularly regret the lack of raw data (and submission to GenBank) allowing a proper analysis of the results. Another disturbing issue results from the fact that a single specimen was tested, preventing quality control, the evaluation of intraspecific diversity, and ensuring specificity.

I hope that Authors can mend these flaws and I heartily encourage resubmission, provided the critical issues raised are satisfactorily answered.

Reviewer 1 ·

Basic reporting

1. The manuscript has room for improvement at the language level: I detected many spelling and grammar mistakes (see examples in lines 69 to 77) that must be corrected.
2. Overall, the presentation of figures and tables could be improved (example: Table 1 contains different character fonts; use of capital letters in column headings is inconsistent); Figures should have higher resolution.
3. Was the raw data deposited on Genbank?

Experimental design

The research is original and primary in the sense that one mitochondrial genome for one species was sequenced and annotated. However, in my opinion this report does not meet the standards of PeerJ because:
NJ (Neighbour-joining) is a distance-based method for building trees that many authors do not consider a true phylogenetic method. NJ has long been replaced by tree-based methods such as Maximum likelihood and Bayesian Inference. I recommend that the authors reconstruct the phylogeny of shown in Figure 4 using ML (e.g. IQ-Tree) and BI (e.g. MrBayes). The authors should also revise the taxa included in the tree shown in Figure 4: I retrieved mitogenomes for 15 species (including subspecies) of Mephitidae on April 6, 2023.
The tree presented on Figure 5 includes some species for which the complete set of PCGs was not available but using NJ as the reconstruction method is presently not appropriate due to the wide availability of ML and BI methods. The polyphyletic state of Spilogale shows that cytB does not recover the true phylogenetic relationships among Mephitidae (as acknowledged by the authors) and this "failed experiment" could be relegated to Supp Material.
I regret to say that cannot consider that the investigation was performed to a high technical standard. Furthermore, the authors merely describe the new mitogenome - comparative mitogenomics within the family could have been presented.

Validity of the findings

My main issue is that the validity of the phylogeny can be questioned due to the use of a distance-based method (NJ).
The new mitogenome expands on existing data for Mephitidae, a family for which mitogenomic data is scarce; however, the report is quite thin in terms of possible analyses and the data included.

Additional comments

The inclusion of stink-skunk from other Southeast Asian islands would greatly increase the scientific impact of this work. I would recommend that the authors invest in generating data that allows for exploring hypotheses on population phylogeographic structure and genetic diversity of the species, update their phylogenetic methods and seek support from a trusted colleague for improving language, style and general presentation of their work.

·

Basic reporting

Please check the Additional Comments

Experimental design

Please check the Additional Comments

Validity of the findings

Please check the Additional Comments

Additional comments

I commend the authors for their work. Overall the manuscript is sound and use clear and unambiguous English language. The objective is relevant and the workflow is fully documented, with methods described with detail and information to replicate. However, some points should be adjusted before Acceptance.

Line 54 – please correct ctyb to cytb

Lines 157 to 165 were fully repeated within the Results section (Lines 204-209 and Lines 215-217). Please avoid this removing them from the Results section.

Lines 223-224 – The authors are saying “Repeated analysis with Maximum Likehood, Maximum Parsimony and UPGMA trees (data not shown) produced identical clusters.” The authors should provide these results within the manuscript or as Supp. Files. This sentence without showing the produced results as well as the parameters used for the three analyses should be avoided/deleted for the final version.

Line 283 – A NJ analysis rather than a NJ tree since the tree is just the result of a distance based method called NJ. Additionally, why the authors choosed a NJ approach rather than a ML or a Bayesian approach for the entire mitogenome? I’m not sure if distance based methods are the best option for a mitogenomic phylogeny. Please explain why the authors choosed to use a NJ approach?

I thank the authors for partially providing the raw data. However, the mitogenomic database, as the second main object of the manuscript after the annotated mitogenome are not present. Imho the authors should also provide this database as well as clarify all the analysis optionw they used to obtain the presented trees, not only the bootstrap number of replicates.

Reviewer 3 ·

Basic reporting

Line(L)74: avoid contractions and mistakes as “It´s”

L84-87: Please reduce the description of the biology of the species and include some more information about the mtDNA of related taxonomic groups, as the work is focused on the mtDNA. Introduce the basic structure of mtDNA in other relates species in order to introduce the subject.

Experimental design

L132: Provide some details on the Barcode Tagged Sequencing (BTSeq).

L150: provide the parameters used in tRNAscan-SE v2.0, ARWEN v1.2 and other programs.

L158: There are now better pgylogenetic methods than Neighbor-Joining. Provide a ML or Bayesian phylogeny. For example, you can use http://www.atgc-montpellier.fr/phyml/
It will be very useful to see if the complete mtDNA would improve current phylogenetic inferences on this group.

L234: “PCR reaction” is redundant

Table 1: the Annealing temperature is the used in PCRs or the predicted for each primer?

Table 4: the information from this table is not very relevant, those values are standard nowadays.

Validity of the findings

L250: Provide some details on the control region of these species. What is the structure and diversity pattern? The CR is often relevant for intra-species studies. Some information here could be useful for future studies. Has it be use before? Can you suggest primer-binding sites? Informative hypervariable regions?

L301-316: this section does not contribute to discuss the findings of this work. Discuss instead the mtDNA and its features.

L318: the Conclusion is more a summary and does not add any relevant information. Please remove it and reinforce the discussion.

---

## Round 0.2 · Minor Revisions

Thank you for the attention you have given to updating the manuscript based on the Reviewer's comments. There are still some minor issues to attend to, which will make your manuscript clearer and more impactful.

·

Basic reporting

no comment

Experimental design

no comment

Validity of the findings

no comment

·

Basic reporting

a. There is still some room for improvement to ensure grammatical correctness. Some examples where the language could be improved include lines 127 – 129, 290, 324 – 325, 363 – the current phrasing makes comprehension difficult.
b. There are some minor adjustments that could be made to the figures and tables. Table 2 seems to contain a different font to the rest of the tables. The phylogenetic trees should be provided at a higher resolution if possible.
c. Although the introduction has been revised to include information about the value of mitogenomic data for this species, there is still no information about the basic composition of mitogenomes of related taxonomic groups. See Gibson et al (Mol Biol Evol, 2005, V 22(2), pp 251: 10.1093/molbev/msi012). It is necessary to describe this in the introduction to provide a point of reference when discussing your results on the structure of your newly assembled mitogenome for Mydaus javanensis.

Experimental design

a. Lines 104 – 109: You state that the aim of the paper is to determine and characterize the complete mitogenome of M. javanensis and discuss how this will fill the identified knowledge gap. It is also worth mentioning that the paper aims to confirm the phylogenetic placement of the species within the family Mephitidae.
b. Lines 202 – 205: Motivate your choice for selecting the cyt b gene over other mitogenomic regions. Has it previously been shown to be informative for interspecific comparisons in these taxa in other studies or was it just the most widely available gene for these taxa? Otherwise simply state that it is widely used as a phylogenetic marker for systematic studies to resolve interspecific divergence.
c. Lines 209 – 216: it looks like the BI analysis for the cyt b gene was conducted in the same manner as for the 13 PCGs. Rather state this to avoid repeating identical parameters in the interest of being concise.

Validity of the findings

a. The conclusion still reads as a summary of the results section. Avoid repeating all the results and rather highlight the most noteworthy ones. State how these findings have addressed the initial research question and helped fill the knowledge gap identified in the introduction and what future research needs to focus on to address any shortcomings of the present study.

Additional comments

a. Consider replacing words already used in the title with new keywords such as phylogenomics, taxonomy, mitogenome assembly, comparative mitogenomics.
b. Table 4 could be easier to read and more valuable if it is presented as a figure. It could even be incorporated with the RSCU in Figure 2. See Wang et al (2022, J Gene, Vol 820: 146232) as an example.
c. I feel that that Figure 3 does not enhance the value of the paper considering the tRNAs of the M. javanensis mitogenome do not show any deviation from what was expected based on mitogenomes from related taxa. Perhaps move this to the supplementary materials.
d. Lines 375 – 380: the nucleotide variations in cyt b should be stated first in the results section with reference made to the supplementary materials S2 – S4. It can then be referred to again in the discussion where it is mentioned that mitogenome sequences should be obtained for Southeast Asian populations to explore the evolutionary patterns further (lines 380 – 385).

---

## Round 0.3 · accepted · Accept

Well done on addressing all the minor comments from the previous review. I am satisfied that the changes you have made have addressed all the previous issues raised. The manuscript is now much more readable and makes a great contribution to the literature - particularly in regards to the biology of Borneo.